# Distribution and Conservation Status of the Mountain Wetlands in the Romanian Carpathians

Claudia Bita-Nicolae 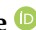

Ecology, Taxonomy & Nature Conservation Department, Institute of Biology Bucharest, Romanian Academy, 060031 Bucharest, Romania; claudia.bita@ibiol.ro

**Abstract:** Mountain wetland habitats are of particular importance because of their biodiversity, their aesthetic and recreational functions, and for providing services to humans (e.g., water for domestic use and livestock). At the same time, these practices can also have significant environment costs, including biodiversity loss and deterioration of water quality. For all their importance, these habitats are not well managed or conserved. The aim of the paper is to study the distribution of two of the most important and vulnerable habitats. The communities of *Cardamino-Montion* and *Cratoneurion commutati* belong, according to the European Red List of Habitats, to the habitats base-poor spring and spring brook (C2.1a) and calcareous spring and spring brook (C2.1b), respectively. This study draws on both original studies and national literature to highlight the characteristic features of mountain wetlands. The main objective of our research is to provide a management framework to facilitate the protection, enhancement and restoration of springs in the Romanian Carpathians and beyond.

**Keywords:** mountain wetland; habitats; *Cardamino-Montion*; *Cratoneurion commutati*; spring vegetation

## 1. Introduction

Mountain wetlands are of particular interest in terms of biodiversity [1,2]. They are located in areas with rich biological heritages [3–6] and are shelter to numerous species, many of which have sensitive populations (i.e., rare and endangered) [7]. However, mountain wetlands are one of the rarest and most fragile habitats [8], threatened by the effects of climate change and overexploitation of water resources [9]. Upland wetlands play an important role in hydrological, ecological and environmental aspects of the watershed [10]. They are spatially restricted in extent, but they also provide many important hydrological and ecological services [11]. For example, wetlands are considered a "hotspot" for global biogeochemical transformation [12]. Moreover, mountain wetlands play a vital role in sequestering terrestrial carbon [13]. Mountain wetland ecosystems are expected to be among the most sensitive to climate change, as their persistence depends on factors directly influenced by climate (i.e., precipitation, snow cover, evaporation) [14,15]. Wetlands are important for nature conservation [6]. Nevertheless, the challenges associated with these mountain wetlands are aggravated by their small size [16,17], which means that many of them cannot be included in wetland inventories [18] and access to them is difficult [19]. Therefore, due to the limited knowledge that scientists have about them and the little information that exists about the characteristics of the biotic communities, the study of wetlands becomes difficult [20]. Moreover, mountain wetland plant communities often enable only a few species from the broader regional species pool to colonize the site, based on each species' dispersal ability, its environmental requirements and the competitive interactions that may facilitate or hinder this [21,22]. These ecosystems have become among the most threatened ecosystems in the world [23,24]. Conservation of these fragile ecosystems is important, particularly in an era of international tourism and climate change [25]. It is essential to refine our knowledge of the vulnerability of biodiversity to climate change in an effort to develop other predictive approaches and to go beyond predictions [26]

in the context that many wetlands are subject to human pressures [27] and that wetland conversion and long-term wetland loss have been over 50% and 87%, respectively, since the beginning of the 18th century [28]. The rapid rate of wetland loss is shocking [29–35].

Despite their importance and climate sensitivity, mountain wetlands tend to be understudied due to a lack of available tools and data [15,25]. There are few studies on the mountain springs in the Western Carpathians [36]; as for the Romanian Carpathians, there are numerous vegetation studies [37,38]. However, there are no studies that consider mountain springs as a whole in either region.

As a result, mountain wetlands located near wetland-protected areas tend to be in better condition compared to remote sites [39]. Plant communities of spring vegetation represent mostly small-scale growth dependent on flowing water [40–42]. In this context, and considering their structural and functional importance highlighted above, the aim of this study is to present an overview of mountain wetlands in the Romanian Carpathians, and their distribution across the whole investigated territory and conservation status. In this paper, information has been gathered from our own database and from the literature. In Romania, there are no comprehensive studies on mountain wetlands and we will fill a gap on this topic.

## 2. Material and Methods

### 2.1. The Area of Study

The Carpathian Mountains are a mountain range belonging to the great central mountain system of Europe. There are numerous areas with karstic and calcareous relief forms, relict glacial relief forms and varied structural and petrographic relief [43–45]. As a rule, the habitats covered by this study are found in the Romanian Carpathians in the middle mountain belt.

The Romanian Carpathians have a temperate mountain climate. In the mid-mountain belt, the climate ranges from 650–800 m to 1850–1900 m and the average temperature is 7 °C. The average rainfall is 800 mm/year [46].

### 2.2. Field Methods

For this study, we used phytosociological relevés according to Central-European School [47,48]. The phytosociological relevés have been carried out in the Romanian Carpathians in an altitude range between 950 m and 1300 m above sea level. These have been subjectively positioned to include most of the observed environmental heterogeneity, but they each cover a single vegetation type. The nomenclature of the syntaxa follows the literature [49].

The angiosperms group taxonomy was performed according to the Euro+ MedPlantBase [50], while The Plant List [51] was used for the currently accepted name of plants and for mosses [52]. We also referred to the national literature to verify taxonomy and names [53,54].

### 2.3. Mapping of the Area

All localities were coded using the UTM (Universal Transverse Mercator) coordinate system, resulting in a 10 × 10 km grid in Romania on the basis of which a database of sites, including localities, was created [55,56]. The UTM system divides the Earth into 60 zones, each of which is 6° of longitude in width. Zone 1 covers longitude 180° to 174° W, and zone numbering increases eastward to Zone 60, which covers longitude 174° E to 180°. The software can visually present the syntaxa's chorology at the scale of 1:6,000,000; the map used presents the multiannual average temperature per year [54].

### 2.4. Structure of Communities

In this study, the networks forming between species were created using VOSviewer [57] by developing and visualizing the networks formed by the taxa of the species found in each analyzed site. VOSviewer is a new software tool that can be used to generate, visualize

and analyze networks that are created between taxa within a habitat. Using VOSviewer, these networks can be visualized at speeds and scales that are not feasible using manual methods or traditional software tools. Clusters have been created according to the close connection between nodes, and they may appear in different colors in each cluster. The node size indicates the co-occurrence or occurrence value and the distance between two nodes represents their approximate relationship.

## 3. Results

*Field Results*

A total of 63 sites with 720 relevés belonging to both the class of *Montio-Cardaminetea* and the order of *Montio-Cardaminetalia* were gathered from the Romanian Carpathians from the literature and from our own database (Tables 1 and 2).

**Table 1.** Sites from the Romanian Carpathians. (Cardamino-Montion alliance).

| ID | Locality | UTM Code | Numbers of Relevés |
|----|----------|----------|--------------------|
| 1 | Piatra Craiului Mt | LL70 | 23 |
| 2 | Bistrita Aurie Valley | LN66 | 10 |
| 3 | Siriu Mt | ML43 | 10 |
| 4 | Postavaru Mt | LL83 | 25 |
| 5 | Fagaras Mt | LL48 | 17 |
| 6 | Sebesului Valley | GR07 | 17 |
| 7 | Tarcu Mt | FR01 | 17 |
| 8 | Iedutului Valley | FS16 | 1 |
| 9 | Sighiselului Valley | FS15 | 2 |
| 10 | Vladeasa Mt | FS86 | 5 |
| 11 | Iadului Valley | FS28 | 6 |
| 12 | Plopis Mt | FT21 | 8 |
| 13 | Cibinului Mt | GR26 | 9 |
| 14 | Draganului Valley | FS39 | 5 |
| 15 | Gurghiului Valley | LM38 | 7 |
| 16 | Defileul Muresului | ES90 | 14 |
| 17 | Zanoaga Mt | LL26 | 4 |
| 18 | Govora Mt | ER99 | 10 |
| 19 | Fagaras Mt | LL40 | 35 |
| 20 | Rodnei Mt | LN35 | 25 |
| 21 | Tarcu-Godeanu Mt | LL65 | 25 |
| 22 | Retezat Mt | FR34 | 5 |
| 23 | Cindrelului Mt | KL76 | 17 |
| 24 | Piatra Craiului Mt | LL71 | 25 |
| 25 | Rodnei Mt | LN36 | 18 |
| 26 | Tarcu-Godeanu Mt | LL67 | 5 |
| 27 | Vladeasa Mt | FS38 | 13 |
| 28 | Retezat Mt | FR36 | 10 |
| 29 | Cindrelului Mt | KL77 | 4 |
| 30 | Bucegi Mt | LL81 | 1 |
| 31 | Maramures Mt | LN17 | 5 |
| 32 | Plopis Mt | FT22 | 5 |
| 33 | Gurghiului Valley | LM40 | 5 |
| 34 | Siriu Mt | ML42 | 4 |
| 35 | Defileul Muresului | ES91 | 10 |
| 36 | Suceava County | MN09 | 4 |
| 37 | Neamt County | MM48 | 4 |
| 38 | Tarcu-Godeanu Mt | LL65 | 6 |
| 39 | Rodnei Mt | LN38 | 10 |
| 40 | Fagaras Mt | LL51 | 1 |

**Table 1.** *Cont.*

| ID | Locality | UTM Code | Numbers of Relevés |
|----|----------|----------|--------------------|
| 41 | Azuga Valley | LL83 | 1 |
| 42 | Bucegi Mt | LL81 | 10 |
| 43 | Nemira Mt | MM51 | 10 |
| 44 | Bihor Mt | FS34 | 5 |
| 45 | Piatra Craiului Mt | LL12 | 7 |
| 46 | Piatra Craiului Mt | LL12 | 3 |
| 47 | Fagaras Mt | LL41 | 8 |
| 48 | Fagaras Mt | LL41 | 15 |
| 49 | Rodnei Mt. | LN42 | 15 |
| 50 | Gurghiului Valley | LM19 | 7 |
| 51 | Gurghiului Valley | LM19 | 8 |
| 52 | Gurghiului Valley | LM38 | 13 |
| 53 | Cindrelului Mt | KN70 | 25 |
| 54 | Maramuresului Mt | FT80 | 10 |

**Table 2.** Cratoneurion commutati alliance.

| ID | Locality | UTM Code | Numbers of Relevés |
|----|----------|----------|--------------------|
| 1 | Retezat Mt | FR55 | 8 |
| 2 | Rodnei Mt. | LN40ll | 11 |
| 3 | Piatra Rea valley | LN23 | 25 |
| 4 | Tarcu-Godeanu Mt. | LL79 | 10 |
| 5 | Bucegi Mt. | LL82 | 15 |
| 6 | Bucegi Mt. | LL82 | 50 |
| 7 | Rodnei Mt. | LN37 | 14 |
| 8 | Rachitisul Mare valley | LN77 | 16 |
| 9 | Maramuresului Mt. | FT95 | 7 |

## 4. Discussions

### 4.1. Distribution of Studied Communities

Wetlands are a particularly valuable ecosystem in the Carpathian region due to their importance in terms of biodiversity conservation and because of the wide variety of unique ecosystem services that are essential for humans [58,59]. These habitats are aquatic habitats, wet meadows, peatlands, riparian vegetation, wet forests, watercourses and subterranean wetlands [55]. Moreover, the generally high-altitude cover and diversity of species in these habitats varies depending on the type of substrate conditions, water chemistry and water temperature [60,61].

Crenic vegetation is found in the wetland in mountainous areas and is mainly composed of species adapted to special habitat conditions, such as constant low water temperature, high air humidity throughout the year and high oxygen saturation [55], and it is usually composed of a mixture of vascular plants, which are more numerous in shaded sites at lower altitudes, and bryophytes in open habitat communities from subalpine to alpine [40,62–64].

Spring species composition reflects the mineral richness of the groundwater, so even a small fluctuation in mineral concentration can lead to vegetation change [65].

Altogether, in the Romanian Carpathians, we found a strong representation of the species that define these communities: *Caltha laeta*, *Cardamine amara*, *Saxifraga stellaris*, *Carex remota* for *Cardamino-Montion* (Figure 1 and *Cratoneuron commutatum*, *Silene pusilla*, *Cratoneuron filicinum* and *Cardamine opizii* for *Cratoneurion commutati* Figure 2).

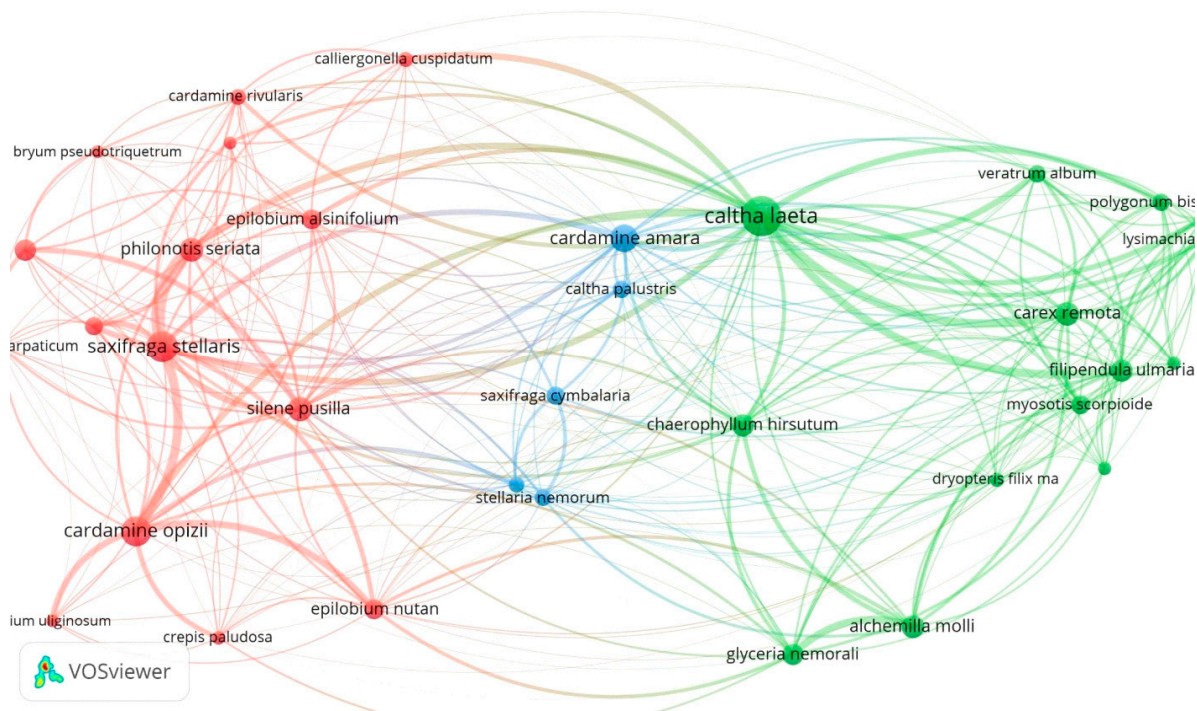

**Figure 1.** *Cardamino-Montion* communities.

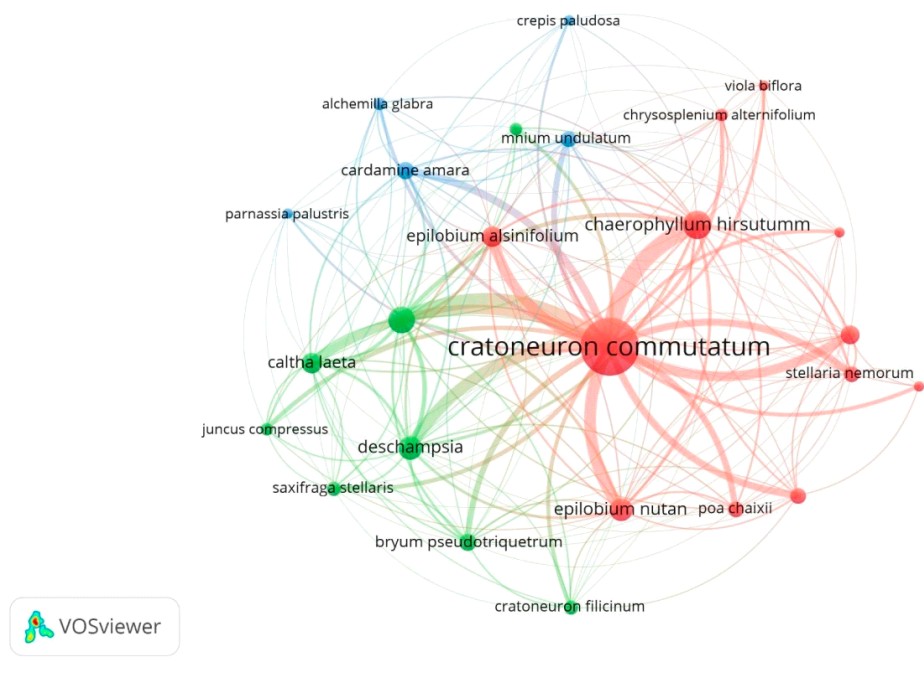

**Figure 2.** *Cratoneurion commutati* communities.

The results showed that the diversity of wetland plant species was high. The abundance of wetland plant species decreases with the increases of elevation and latitude, and increases with the increase of longitude [66]. Wetland hydrological conditions, soil microtopography and microbial activity amplify the contribution of soil properties to changes in plant biomass, cover and diversity [67].

Despite their importance, springs are much less studied than other aquatic ecosystems. They also are insufficiently covered by protective legislation, often resulting in the destruction of their natural habitat [68].

The studied communities were from 63 sites in the Romanian Carpathians belonging to the *Montio-Cardaminetea* class and the *Montio-Cardaminetalia* order.

The *Montio-Cardaminetea* class groups habitats from the edge of springs and cold streams on the mountain, subalpine and alpine superior. The floristic composition is determined by the constant limits of uninterrupted water flow and temperature throughout the growing season. The formation and maintenance of fontinal communities is conditioned by the rapid flow of streams, which enriches their oxygen content and rarely exceeds +5C. Over time, the limited nature of this ecological alliance selected by selective integration a well-defined complex of species, of which the fontinal cenoses are endowed with a remarkable floristic conservatism whenever erosive processes interfere with the canvas of the springs. It includes the montane fontinal vegetation of Europe, which contains often floristic features according to the geological substratum, siliceous, or calcareous, where they develop. Among the characteristic species present in the Romanian Carpathians we examine *Cardamine amara*, *Caltha laeta*, *Epilobium nutans*, *E. alsinifolium*, *Saxifraga stellaris* and *Bryum pseudotriquetrum*.

The paper brings to the fore the two alliances of the *Montio-Cardaminetalia* order and *Montio-Cardaminetea* class: *Cardamino-Montion* and *Cratoneurion commutati*.

The communities of *Cardamino-Montion* are numerous, spread almost throughout the entire area of the Romanian Carpathian, and represent herbaceous vegetation on alpine river banks and the vegetation of cold oligotrophic water (with low pH) springs. However, the habitat is very widespread in Europe as well. The alliance contains the vegetation of the streams in the subalpine and alpine belts of mountains of the Carpathians from the acid till neutral substratum (pH = 4–6.8). The water warms up easily because of the low amounts of water discharged by the springs and because of the dark color of the bryophytes [69]. The habitat comprises moisture-loving vegetation along high mountain streams (alpine and subalpine belts above 1800 m altitude) and the siliceous substrate is wet and stony. Due to the very late thaw, the growing season is very short (about two months per year).

On the other hand, *Cratoneurion communities* are less common, requiring certain geographic conditions, such as limestone rocks. Thus, the habitat can be found around springs in rocky mountainous areas, where there are extensive pads of moss and mainly populated by the characteristic species of *Cratoneurion*.

This alliance was defined principally by abiotic attributes—lime-rich spring communities [69]—and it contains the spring-growing phytocoenosis consisting of basiphilous components developed close by the streams and springs on the calcareous substrate.

The two communities studied are found throughout the Romanian Carpathians (Figure 3) and this is a good sign for their ecology.

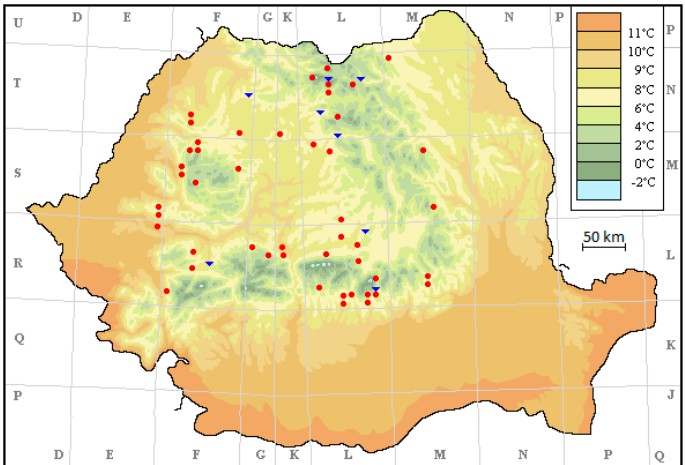

**Figure 3.** The distribution of studied communities. ● *Cardamino-Montion* communities; ▼ *Cratoneurion commutati* communities.

### 4.2. Conservation and Management

According to EUNIS habitat classification [70], *Cardamino-Montion* communities belong to the habitat base-poor spring and spring brook (C2.1a), while *Cratoneurion commutati* belongs to the habitat calcareous spring and spring brook (C2.1b) (Table 3).

**Table 3.** Conservation status of studied habitats.

| Habitat | Order | Red List Habitat Type | Threat Status Europe | Threat Status EU | Annex I Habitat Type |
|---|---|---|---|---|---|
| Base-poor spring and spring brook | Cardamino-Montion | RLC2.1a | Near Threatened | Vulnerable | 7220 Petrifying springs with tufa formation (Cratoneurion) |
| Calcareous spring and spring brook | Cratoneurion commutati | RLC2.1b | Vulnerable | Vulnerable | 7220 Petrifying springs with tufa formation (Cratoneurion) |

Petrifying springs with tufa formation (*Cratoneurion*) constitute a priority habitat (7220) under Annex I of the European Union Habitats Directive (92/43/EEC) owing to their ecological significance, vulnerability and small spatial extent [71] (Table 3).

The communities of springs have severely declined in many countries in Europe. Calcareous springs, spring brooks and tufa cascades have undergone severe losses in quantity in many countries historically and also in the recent past, but they still have a very large distribution range.

According to the European Red List of Habitats [72], indicators of good quality that can be inferred from this study include a high cover of moss and specialized vascular plants and a low cover of tall grasses and encroaching shrubs. On the other hand, low anthropogenic influence (e.g., drainage, water exploitation, forestry) in and around springs and catchments is also observed.

Protecting natural hydrology and limiting contamination are the main solutions for the conservation of springs and their surroundings. Mountain springs are small-scale habitats, so their vegetation is sensitive to change. Representative spring sites should be legally protected. However, specific schemes for the management and restoration of spring biodiversity need to be developed.

### 5. Concussions

The two habitats are well represented in the Romanian Carpathians. Considering both their importance in the local and regional ecological balance and their vulnerability, it is essential to know that these habitats are affected to a small extent by anthropogenic pressure.

To reduce the threat to freshwater ecosystems at both local and regional scales, there are many important actions in their management. However, this is often a social, political and financial challenge rather than a purely technical one [71].

Protected areas are crucial for ecosystem conservation [41]. Protected areas aim to promote in situ conservation strategies for threatened habitats and species by creating a network of managers and scientific experts to support capacity building, management and policy actions [73].

Worldwide biodiversity loss is one of the most important threats confronting the planet. Addressing this problem requires a wide variety of efforts. One step that conservationists can take is to make sure they are framing biodiversity loss in ways that communicate effectively to as many stakeholders as possible [74].

**Funding:** This research was funded by the project RO1567-IBB01/2022 of the Romanian Academy.

**Institutional Review Board Statement:** Not applicable.

**Informed Consent Statement:** Not applicable.

**Data Availability Statement:** Publicly available datasets and personal data were analyzed in this study. The national literature data are found according to references [38].

**Acknowledgments:** Sorin Stefanut is acknowledged for assisting in data analysis. The author would also like to thank the editors and reviewers for their suggestions that have significantly improved the quality of this paper.

**Conflicts of Interest:** The author declare no conflict of interest.

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
