# Peer review of "Distribution and Conservation Status of the Mountain Wetlands in the Romanian Carpathians"

_sustainability, doi:10.3390/su142416672_

Round 1
Reviewer 1 Report
Comments on manuscript titled “Distribution and Conservation Status of the Mountain Wetlands in the Romanian Carpathians” by Claudia Bita-Nicolae
(Manuscript Number: sustainability-1981156)
Mountain wetlands are important due to their biodiversity, aesthetic and recreational function and provide services to humans, such as supply water for domestic use and livestock. Using mountain wetlands have significant costs to the environment, including biodiversity loss and deterioration of water quality. For all their importance these habitats are not so well managed and conserved. The aim of the paper is to study the distribution of two of the most important and vulnerable such habitats. The communities of Cardamino-Montion and Cratoneurion commutati belong, according to the European Red List of Habitats, to 2.1a Base-poor spring and spring brook and C2.1b Calcareous spring and spring brook, respectively. The study draws on both our own studies and national literature to highlight the characteristic features of mountain wetlands. The objective of the research is to provide a management framework to facilitate the protection, enhancement and restoration of springs in the Romanian Carpathians and beyond.
The conclusions of this manuscript are believable. I recommend that the manuscript be accepted for publication with major revision in Sustainability.
Some suggestions:
1) The English should be improved.
2) The methods should be explained in more detail. For example, 2.4. Structure of communities. In this study, the networks forming between species were created using VOSviewer [54].
3) How many references used for analyzing, and how the references been selected should be given clearly.
4) In the section of discussions, how the conserving states of the habitats should be discussed in detail.
Author Response
Thanks very much for taking your time to review this manuscript

Reviewer 2 Report
Distribution and Conservation Status of the Mountain Wet- 2 lands in the Romanian Carpathians is a good study. It has a scope but I have some suggestions, incorporation of which will definitely enhance the readability and quality of manuscript.
Introduction has been divided in unnecessary paragraphs, try to merge the paragraphs up to 2-3 only.
A map with well labelled coordinates showing the location of study area should be added.
Materials and methods need a revision. It should be well elaborated so that it can be replicable.
On which basis communities were identified?
I endorse the publication of ms after minor revision
Author Response

(The authors gave the same response as above.)

Reviewer 3 Report
The manuscript "Distribution and conservation status of the mountain wetlands in the Romanian Carpathians" by Claudia Bita-Nicolae tries to analyze the state of conservation and management of mountain wetlands in the Carpathian mountains of Romania.
The manuscript lacks to the necessary organization of ideas to be transmitted. The objectives of the work are not clear either, being those exposed much broader than those that it tries to answer. The material and methods are also very poor, and it is not clear how the state of conservation and management of these ecosystems is evaluated. Very general ideas that contribute very little to the general knowledge of these ecosystems in the Carpathian Mountains of Romania.
The presentation of the results is deficient, without the necessary text presenting the obtained results. The figures are repeated (there are two figures 1 and two figure 2) and some are not even cited. What aspect of the community structure is measured/evaluated? Finally, the discussion section is also very poor, with few bibliographical references, only five.
It is also necessary to review the English by a native.
Author Response

(The authors gave the same response as above.)

Round 2
Reviewer 1 Report
All the required review are performed.
However, the serial numbers of refferences should be ranked again correctly. There are two references ranked No. 26.
Author Response
Dear Reviewer,
I appreciate the effort to thoroughly revise this manuscript. I have corrected the error that appeared and also revised the English language once again. Many thanks!